
# Dipolar condensed atomic mixtures and miscibility under rotation

**Lauro Tomio[1⋆], Ramavarmaraja Kishor Kumar[1,2] and Arnaldo Gammal[3]**

**1** Instituto de Física Teórica, Universidade Estadual Paulista, 01140-070 São Paulo, SP, Brazil.
**2** Department of Physics, Centre for Quantum Science, and Dodd-Walls Centre for Photonic and Quantum Technologies, University of Otago, Dunedin 9054, New Zealand.
**3** Instituto de Física, Universidade de São Paulo, 05508-090 São Paulo, Brazil.

⋆ lauro.tomio@gmail.com

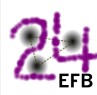

## Abstract

By considering symmetric- and asymmetric-dipolar coupled mixtures (with dysprosium and erbium isotopes), we report a study on relevant anisotropic effects, related to spatial separation and miscibility, due to dipole-dipole interactions (DDIs) in rotating binary dipolar Bose-Einstein condensates. The binary mixtures are kept in strong pancake-like traps, with repulsive two-body interactions modeled by an effective two-dimensional (2D) coupled Gross-Pitaevskii equation. The DDI are tuned from repulsive to attractive by varying the dipole polarization angle. A clear spatial separation is verified in the densities for attractive DDIs, being angular for symmetric mixtures and radial for asymmetric ones. Also relevant is the mass-imbalance sensibility observed by the vortex-patterns in symmetric- and asymmetric-dipolar mixtures. In an extension of this study, here we show how the rotational properties and spatial separation of these dipolar mixture are affected by a quartic term added to the harmonic trap of one of the components.

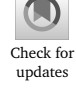
## 1 Dipolar Bose-Einstein condensate - Introduction

The experimental realization of Bose-Einstein condensation in chromium ($^{52}$Cr) atoms has opened the new research direction called dipolar quantum gases [1]. Following the condensation in $^{52}$Cr, many subsequent studies have been carried out by different experimental groups on fermionic and bosonic properties of strongly dipolar ultracold gases, such as with dysprosium (Dy) and erbium (Er) (see [2] and references therein). More recently, the elementary excitations spectrum of $^{164}$Dy and $^{166}$Er dipolar Bose gases were analyzed in Ref. [3], by considering three-dimensional (3D) anisotropic traps across the superfluid-supersolid phase transition. The recent investigations in ultracold laboratories with two-component dipolar Bose-Einstein Condensates (BEC), on stability and miscibility properties, became quite interesting

due to the number of control parameters that can be explored in new experimental setups. The parameters which can be controlled are given by the strengths of dipoles, the number of atoms in each component, the inter- and intra-species scattering lengths, as well as confining trap geometries. The stability and pattern formation have been studied in Ref. [4], by considering dipolar-dipolar interactions (DDIs) in two-component dipolar BEC systems. Rotational properties of two-component dipolar BEC in concentrically coupled annular traps were also studied in Ref. [5], by assuming a mixture with only one dipolar component.

Following previous studies with rotating binary dipolar mixtures and their miscibility properties [6–8], the miscible-immiscible transition (MIT) of the dipolar mixtures with $^{162,164}$Dy and $^{168}$Er were also recently studied by us in Ref. [9]. For these coupled dipolar systems, the miscible-immiscible stable conditions were analyzed within a full 3D formalism, by considering repulsive contact interactions, within pancake- and cigar-type trap configurations. The rotational properties and vortex-lattice pattern structures of these dipolar mixtures were further investigated by us in Refs. [10–12], by changing the inter- to intra-species scattering lengths, as well as the polarization angles of the dipoles. Among the observed characteristics of these strong dipolar binary systems, relevant for further investigations are the possibilities to alter the effective time-averaged DDI from repulsive to attractive, by tuning the polarization angle $\varphi$ of both interacting dipoles from zero to 90°, respectively. In Ref. [11], vortex pattern structures were studied by considering rotating binary mixtures confined by squared optical lattices, whereas in Ref. [12], by tuning $\varphi$, our investigation was mainly concerned with rotational properties together with spatial separations of the binary mixtures.

Motivated by the above mentioned studies, considering that the interplay between DDIs and contact interactions can bring us different interesting effects in the MIT, showing richer vortex-lattice structures in rotating binary dipolar systems, in the present contribution we are also reporting some new results obtained for the properties of dipolar mixtures confined by a strongly pancake-like two-dimensional (2D) rotating harmonic trap. The effect of a weak quartic perturbation in the $(x, y)$ plane, applied to the first dipolar component, is studied by tuning the polarization angles of the dipoles together with the contact inter-species interactions.

Next section, the model formalism is presented with our parametrization and numerical procedure. In section 3, after an analysis of the main results for symmetric and asymmetric binary dipolar mixtures confined by strong pancake-like harmonic traps, we present new results obtained when considering the effect of a weak quartic perturbation added to the harmonic trap of one of the components. Finally, a summary with our conclusions is given in section 4.

## 2 Model formalism, parametrization and numerical approach

### 2.1 Model formalism

The coupled dipolar system with condensed two atomic species $i = 1, 2$, with the respective masses $m_i$ (with $m_1 \geq m_2$) are assumed to be confined in strongly pancake-shaped harmonic traps, with fixed aspect ratios, such that $\lambda = \omega_{i,z}/\omega_{i,\perp} = 20$ for both species $i = 1, 2$, where $\omega_{i,z}$ and $\omega_{i,\perp}$ are, respectively, the longitudinal and transverse trap frequencies. The coupled Gross-Pitaevskii (GP) equation is cast in a dimensionless format, with energy and length units given, respectively, by $\hbar\omega_{1,\perp}$ and $l_\perp \equiv \sqrt{\hbar/(m_1\omega_{1,\perp})}$. Correspondingly, the space and time variables are given in units of $l_\perp$ and $1/\omega_1$, respectively, such that $\mathbf{r} \to l_\perp \mathbf{r}$ and $t \to \tau/\omega_1$. Within these units, and by adjusting both trap frequencies such that $m_2\omega_{2,\perp}^2 = m_1\omega_{1,\perp}^2$, the dimensionless external 3D trap potential for each one of the species $i$ can be written as $V_{3D,i}(\mathbf{r}) \equiv V_i(x, y) + \frac{1}{2}\lambda^2 z^2$. On the miscibility conditions for binary trapped dipolar systems, more details and discussion can be found in Refs. [10–12]. Large values for $\lambda$ allow us to

reduce to 2D the original 3D formalism by considering the usual factorization of the 3D wave function as $\psi_i(x, y, \tau)\chi_i(z)$, where $\chi_i(z) \equiv (\lambda/\pi)^{1/4} e^{-\lambda z^2/2}$. The two-body contact interactions related to the scattering lengths $a_{ij}$, and DDI parameters are defined as [12]

$$g_{ij} \equiv \sqrt{2\pi\lambda}\frac{m_1 a_{ij} N_j}{m_{ij} l_\perp}, \;\; \mathrm{d}_{ij} = \frac{N_j}{4\pi}\frac{\mu_0\mu_i\mu_j}{\hbar\omega_1 \, l_\perp^3}, \;\; a_{ii}^{(d)} \equiv \frac{1}{12\pi}\frac{m_i}{m_1}\frac{\mu_0\mu_i^2}{\hbar\omega_1 l_\perp^2}, \;\; a_{12}^{(d)} = a_{21}^{(d)} = \frac{1}{12\pi}\frac{\mu_0\mu_1\mu_2}{\hbar\omega_1 l_\perp^2},$$
(1)

where $i, j = 1, 2$, with $N_j$ being the number of atoms and $m_{ij} = m_i m_j/(m_i + m_j)$ the reduced mass of the species $i$ and $j$. In our numerical analysis, the length unit will be assumed being $l_\perp = 1\mu m \approx 1.89 \times 10^4 a_0$, with $a_0$ being the Bohr radius. The corresponding 2D coupled GP equation for the two components $\psi_i \equiv \psi_i(x, y, \tau)$ of the total wave function can be written as

$$i\frac{\partial \psi_i}{\partial \tau} = \left[\frac{-m_1}{2m_i}\nabla_{2D}^2 + V_i(x, y) - \Omega L_z + \sum_{j=1,2} g_{ij}|\psi_j|^2 + \sum_{j=1,2}\mathrm{d}_{ij}\int dx'dy' V^{(d)}(x-x', y-y')|\psi_j'|^2\right]\psi_i, \quad (2)$$

where $\nabla_{2D}^2 \equiv \frac{\partial^2}{\partial x^2} + \frac{\partial^2}{\partial y^2}$, $\psi_i' \equiv \psi_i(x', y', \tau)$, with $V^{(d)}(x, y)$ being the reduced 2D expression for the DDI. The 2D confining potential $V_i(x, y)$ is assumed to be harmonic for both components, as in Ref. [12]. However, in the present contribution we are providing an extension to our study reported in Ref. [12], by examining the effect, on the pattern distribution and spatial separation of the dipolar mixture, of a quartic term applied to one of the components, which we define as the more-massive one. So, the trap is given by

$$V_i(x, y) \equiv V_i(\rho) \equiv \frac{\rho^2}{2} + \kappa_i \rho^4, \;\; \text{where } \rho \equiv \sqrt{x^2 + y^2}, \;\; \kappa_2 = 0, \quad (3)$$

with $\kappa_1 \equiv \kappa$ being a dimensionless positive parameter (in principle, assumed to be small), which increases the trap confinement of the more massive component. Experimentally, the quartic potential together with harmonic trap can be created by using far-detuned laser beam propagating along the axis of the trap, perpendicular to the $(x, y)$ plane. So, the width and strength of the quartic trap can be controlled, respectively, by the width and amplitude of the blue-detuned Gaussian laser beam. More details can be found in the reference [13], where experiments with quartic trap in BEC are discussed. Each component of the wave function is assumed normalized to one, $\int_{-\infty}^\infty dx dy|\psi_i|^2 = 1$. In Eq.(2), $L_z$ is the angular momentum operator (in units of $\hbar$), with $\Omega$ being the corresponding rotation parameter (in units of $\omega_1$), which is assumed to be common for both components.

The 2D DDI presented in the integrand of the second term shown in Eq. (2) can be expressed in the 2D momentum space as the combination of two terms, by considering the orientations of the dipoles $\varphi$ and the projection of the corresponding Fourier transformed $V^{(d)}(x, y)$. One term is perpendicular, with the other parallel to the direction of the dipole inclinations, as described in Refs. [7,8]. By generalizing the description to a polarization field rotating in the $(x, y)$ plane, the two terms can be combined according to the dipole orientations $\varphi$, with the total 2D momentum-space DDI given by [12]

$$\widetilde{V}^{(d)}(k_x, k_y) = \frac{3\cos^2\varphi - 1}{2}\left[2 - 3\sqrt{\frac{\pi}{2\lambda}}k_\rho \exp\left(\frac{k_\rho^2}{2\lambda}\right)\mathrm{erfc}\left(\frac{k_\rho}{\sqrt{2\lambda}}\right)\right] \equiv V_\varphi(k_\rho), \quad (4)$$

where $k_\rho^2 \equiv k_x^2 + k_y^2$, with erfc(x) being the complementary error function of $x$. The 2D configuration-space effective DDI is obtained by applying the convolution theorem in Eq. (2), performing the inverse 2D Fourier-transform for the product of the DDI and density, such that $\int dx'dy' V^{(d)}(x-x', y-y')|\psi_j'|^2 = \mathcal{F}_{2D}^{-1}\left[\widetilde{V}^{(d)}(k_x, k_y)\widetilde{n}_j(k_x, k_y)\right]$. From Eq. (4), one should notice that such momentum-space Fourier transform of the dipole-dipole potential changes the sign at some particular large momentum $k_\rho$. However, after applying the convolution theorem with the inverse Fourier transform (by integrating the momentum variables), the

corresponding coordinate-space interaction has a definite value, as in the 3D case, which is positive for $\varphi \leq \varphi_M$, and negative for $90° \geq \varphi > \varphi_M$, where $\varphi_M \approx 54.7°$ is the so-called "magic angle", in which the DDI is canceled out.

## 2.2 Parametrization and numerical approach

The two binary mixtures ($^{164}$Dy-$^{162}$Dy and $^{168}$Er-$^{164}$Dy) that we are investigating are called, respectively, "symmetric" and "asymmetric" ones; where these terms are related to the dipolar symmetry of the condensed atoms. The corresponding magnetic dipole moments of the three species are the following: $\mu = 10\mu_B$ for $^{162,164}$Dy, and $\mu = 7\mu_B$ for $^{168}$Er. So, by considering the definitions given in (1), the strengths of the DDI are $a_{ij}^{(d)} = 131\,a_0$ ($i,j = 1,2$), for the symmetric-dipolar mixture $^{164}$Dy-$^{162}$Dy; and $a_{11}^{(d)} = 66\,a_0$, $a_{22}^{(d)} = 131\,a_0$ and $a_{12}^{(d)} = a_{21}^{(d)} = 94\,a_0$, for the $^{168}$Er-$^{164}$Dy mixture. In all the cases, we assume the number of atoms for both species are identical and fixed at $N_1 = N_2 = 5000$. The number of atoms are reduced in relation to the ones used in Ref. [12], in view of our present aim and numerical convenience. For symmetric-dipolar mixture ($\mu_1 = \mu_2$) we have $d_{12} = d_{11} = d_{22}$. In the case of contact interactions, we should consider enough large repulsive scattering lengths in view of our stability requirements. We fix both intra-species contact interactions at $a_{11} = a_{22} = 50a_0$, remaining the inter-species one to be explored by varying the ratio parameter $\delta \equiv a_{12}/a_{11}$. Once selected the polarization angle and $\delta$ as the appropriate parameters to alter the miscibility properties of a mixture, we fix other parameters guided by possible realistic settings and stability requirements. For the present approach, we choose $\Omega = 0.75$ for the rotation frequency parameter, larger than the one used in Ref. [12] ($\Omega = 0.6$), in order to improve the observation of vortex-pattern structures and spatial separation.

For the numerical approach to solve the GP formalism (2), the split-step Crank-Nicolson method [14,15] is applied, combined with a standard method for evaluating DDI integrals in momentum space, as described in Ref. [12]. In the search for stable solutions, the numerical simulations were carried out in imaginary time on a grid with a maximum of 464 points in both $x - y$ directions, with spatial and time steps $\Delta x = \Delta y = 0.05$ and $\Delta t = 0.0005$, respectively. In this approach, both wave-function components are renormalized to one at each time step.

# 3 Symmetric- and asymmetric-dipolar mixtures - Results

## 3.1 Dipolar mixtures confined by identical harmonic pancake-like traps

We focus our study in the two coupled mixtures given by $^{168}$Er-$^{164}$Dy and $^{164}$Dy-$^{162}$Dy, motivated by recent experimental studies with dipolar BEC systems. In our investigation, we have considered harmonic strongly pancake-like trap, as detailed in Ref. [12]. First, a detailed analysis of ground state and stability properties was performed in the absence of rotation. In this respect, we understand that our theoretical predictions can be helpful in verifying miscibility properties in on-going experiments under different anisotropic trap configurations. The stability regime was verified for $^{168}$Er-$^{164}$Dy and $^{164}$Dy-$^{162}$Dy mixtures considering the fraction number of atoms for each species as functions of the trap-aspect ratio $\lambda$. From the MIT conditions for homogeneous coupled systems confined in hard-wall barriers, one can observe that the miscibility remains unaffected by the dipolar interactions. In order to estimate the miscibility for non-homogeneous confined binary mixtures, a relevant parameter $\eta$ was defined in Ref. [9], by integrating the square-root of the product of the two-component densities, given by $\eta = \int \sqrt{|\phi_1|^2 |\phi_2|^2}\, d\mathbf{x}$, which varies from $\eta = 0$ (complete immiscible mixtures) to $\eta = 1$ (for complete miscible mixtures). This parameter is found appropriate for a quantitative estimate

of the overlap between the two densities of the coupled system. By considering the natural properties of the mixed elements, the two mixtures, we notice that $^{168}$Er-$^{164}$Dy and $^{164}$Dy-$^{162}$Dy have quite different miscibility behaviors, with $^{164}$Dy-$^{162}$Dy being almost completely miscible ($\eta = 0.99$) and $^{168}$Er-$^{164}$Dy partially miscible ($\eta = 0.77$), when the other parameters (trap-aspect ratio and number of atoms) are fixed to the same values. Such behavior is clearly due to a mass-imbalance effect, as discussed in Ref. [12], which plays a relevant role in the inter-species dipolar strength when compared with the intra-species one.

The two binary mixtures ($^{164}$Dy-$^{162}$Dy and $^{168}$Er-$^{164}$Dy) are considered in a rotating frame within quasi-2D settings. Owing to the different miscibility properties, quite distinct vortex patterns are observed between the symmetric and asymmetric mixtures. For the dipolar symmetric mixture, $^{164}$Dy-$^{162}$Dy, we observe the following lattice patterns: triangular, square-shaped, rectangular-shaped, double core, striped, and with domain walls. For the dipolar asymmetric mixture, $^{168}$Er-$^{164}$Dy, we notice triangular, square-shaped, and circular pattern lattices. Further, to analyze the anisotropic properties of dipolar interactions, the polarization angle $\varphi$ of the dipoles was modified with the dipolar interactions being tuned from repulsive to attractive. With the dipoles of the two species polarized in the same direction, perpendicular to the direction of the dipole alignment ($\varphi = 0$), the DDI is repulsive. By tuning the polarization angle $\varphi$ from zero to 90° the DDI changes from repulsive to fully attractive, with the DDI being canceled for $\varphi = \varphi_M \approx 54.7°$. The miscibility of the condensed mixture is mainly affected by the inter-species interactions; with the vortex-pattern structures being related to combined effects due to inter- and intra-species interactions, with the vortex-pattern formations obtained with $\varphi = 0$ surviving approximately up to $\varphi \approx \varphi_M$. Complete spatial separation between the two-component densities under rotation is verified for large $\varphi$, when the DDI is attractive. As verified, half-space angular separations occur in symmetric-dipolar cases, represented by $^{164}$Dy-$^{162}$Dy; whereas radial-space separations occur for asymmetric-dipolar cases, represented by $^{168}$Er-$^{164}$Dy. Another quite relevant result obtained in Ref. [12] is the observed effect of the mass-asymmetry in the miscibility and vortex-pattern structures. The particular mass-imbalance sensitivity can better be appreciated in the symmetric-dipolar mixture $^{164}$Dy-$^{162}$Dy for $\delta = 1$, when all the differences between the density patterns should be attributed to the small mass-asymmetry.

Next, we report new results with the trap interaction as given by Eq. (3), with a quartic term added to the harmonic interaction of the more-massive component of both two mixtures.

## 3.2 Dipolar symmetric $^{164}$Dy-$^{162}$Dy mixture, with a quartic trap applied to $^{164}$Dy

As discussed above, being dipolar symmetric, with $a_{11} = a_{22} = 131a_0$, this $^{164}$Dy-$^{162}$Dy BEC mixture exposes more miscible properties. As verified in Ref. [12], this mixture in the rotating harmonic trap [with $\kappa_i = 0$ in Eq. (3)] shows triangular, squared, rectangular-shaped, double core, striped, and with domain wall vortex lattices regarding the ratio between inter- and intra-species contact interaction. Also, this mixture shows complete spatial separation at large polarization angles, where the DDI is purely attractive. By modifying the external confinement of one of the components, we can introduce some external asymmetry to the mixture. So, in this contribution, for this binary system we start by adding a very weak quartic term in the first component of the mixture, in order to analyze the miscibility and complete spatial separation of the coupled system. We consider two different miscible cases, with $\delta = 1$ and 1.45. For these particular cases, striped and domain wall vortex structures are observed [12], respectively, when both species are under identical rotating harmonic pancake-like traps, with $\lambda = 20$ and $\Omega = 0.6$. By adding a quartic term to the trap, as explained in section 2, we have also reduced the number of atoms to $N_i = 5000$ and increased the frequency to $\Omega = 0.75$ in order to improve our observation on the corresponding rotational structure and spatial separation. In this case, ring lattice structures can be verified, also verified even for single component

BECs. From our results, in this communication, we select three different orientation angles of the dipoles, given by $\varphi = 0°$, $45°$ and $90°$, in which the first ($\varphi = 0°$) provides complete repulsive DDI, the second ($\varphi = 45°$) is weakly repulsive, with the DDI of the third $\varphi = 90°$ being complete attractive. As shown by our results presented in Fig. 1, the small weak quartic perturbation in the trap, given by $\kappa = 0.05$, induces radial spatial separations between the condensate densities, displaying ring lattice structure in the second component as shown in the panels (a) to (e) of Fig. 1. The quartic trap term added to the first component makes the first component more confined than the second one. So the second component becomes radially phase-separated, changing the previous patterns observed in Ref. [12] for $\kappa = 0$. Such similar behavior for non-dipolar mixtures was also analyzed theoretically recently in Ref. [16].

To improve our understanding of the phase separation and the effect of the added quartic trap, we studied the dipolar binary system by increasing the strength $\kappa$. We observed that, for $\kappa \geq 0.1$, with large repulsive inter-species interaction $\delta = 1.45$, the spatial phase separations of the densities change completely from the previous angular to radial ones. This behavior is indicated in Fig. 2, where the phase-separated case, displayed for $\varphi = 90°$ with $\kappa = 0.05$, is being compared with the $\kappa = 0.08$ case. So, when $\kappa \geq 0.1$, only radial spatial separation can be observed in the binary mixture.

### 3.3 Dipolar asymmetric $^{168}$Er-$^{164}$Dy mixture, with a quartic trap applied to $^{168}$Er

In this subsection we consider the dipolar asymmetric $^{168}$Er-$^{164}$Dy BEC system, to study the effect of a quartic dipolar trap applied to the first component ($^{168}$Er) of the mixture. As reported in Ref. [12] for this asymmetric-dipolar case, when $\kappa_i = 0$ in the rotating confining harmonic trap given by Eq. (3), one should observe triangular, square-shaped, and circular lattices, by varying the inter-species interaction. The interplay between the inter-species repulsive character, shown by increasing $\delta$, together with the attractive role of the DDI as the polarization angle is increased, have shown radial density distributions for the binary mixture such that for $\varphi = 0°$ and $\delta \geq 1$ the $^{168}$Er element is at the center of the mixture (surrounded by $^{164}$Dy), moving to the external part when the dipolar interaction becomes more attractive, with $\varphi = 90°$, with an exchange of the densities.

Now, with the present study, by increasing the external trap interaction with the quartic term, as given by Eq. (3) with $\kappa = 0.05$, one can already observe some differences in the pattern distribution of the vortices of both mixtures, as shown in Fig. 3. However, one should notice that, for the complete spatial separation that occurs for $\varphi = 90°$, the position of both elements remains as in the case that $\kappa = 0$, implying that the added quartic term is not enough to change the position of the density distributions. More interesting behavior can be observed by increasing the strength of the quartic term, as verified in the Fig. 4, by considering $\varphi = 90°$ with $\delta = 1.45$. In the left panels of this figure, we consider $\kappa = 0.25$, where we can verify that the previous radial distribution of the densities is being modified with the radius of the first component being reduced. With $\kappa = 1$, we finally obtain a radial spatial separation in which the $^{168}$Er condensate is occupying the center, surrounded by the $^{164}$Dy condensate. The densities of the two components interchange their positions in relation to the case that $\kappa = 0$, due to the quartic trap term, which is dominating the confinement of the $^{168}$Er condensate.

## 4 Summary

By considering the symmetric- and asymmetric-dipolar coupled mixtures, respectively given by $^{164}$Dy-$^{162}$Dy and $^{168}$Er-$^{164}$Dy, in this communication we have first discussed rotational prop-

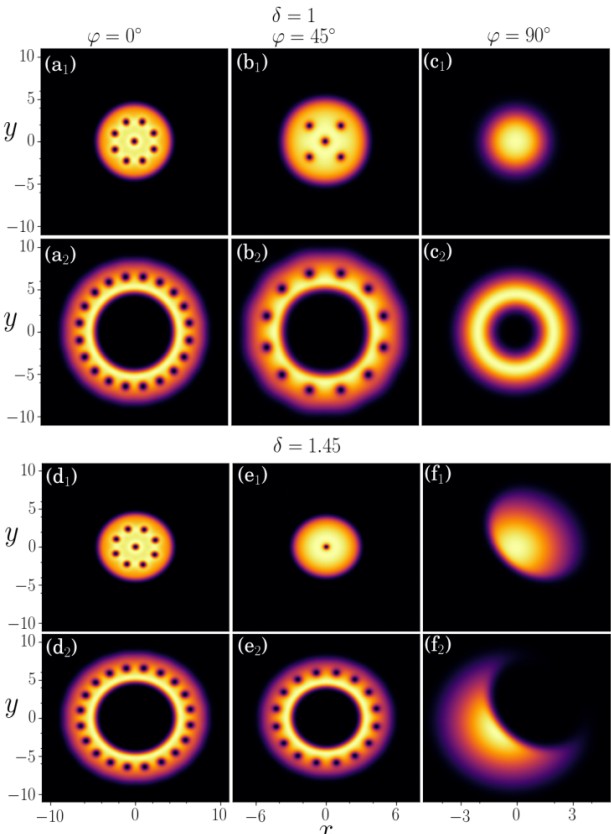

Figure 1: 2D Dipolar density patterns, $|\psi_j|^2$, where $j = 1$ is for $^{164}$Dy and $j = 2$ for $^{162}$Dy, are shown for $\delta = 1$ [$(a_j)$ to $(c_j)$] and $\delta = 1.45$ [$(d_j)$ to $(f_j)$]. The dipole polarization angles ($\varphi = 0°$, $45°$, $90°$) are indicated at the top of each column, with the $^{164}$Dy component having the addition of a quartic trap with $\kappa = 0.05$. The other parameters are: $N_{j=1,2} = 5000$, $\lambda = 20$, $a_{11} = a_{22} = 50a_0$ and $\Omega = 0.75$.

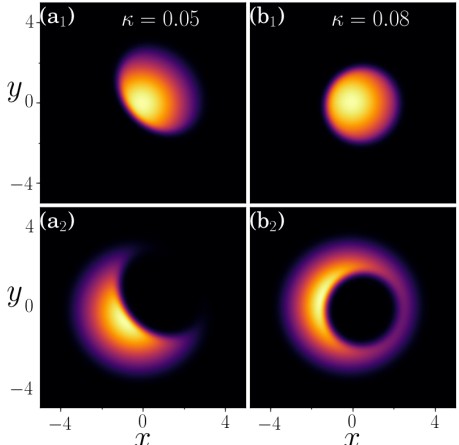

Figure 2: 2D Dipolar density patterns, $|\psi_j|^2$, where $j = 1$ is for $^{164}$Dy and $j = 2$ for $^{162}$Dy, are shown for $\varphi = 90°$ and $\delta = 1.45$. The quartic trap added to component 1 is such that $\kappa = 0.05$ in the left panel and $0.08$ in the right panel. As in Fig. 1, the other parameters are: $N_{j=1,2} = 5000$, $\lambda = 20$, $a_{11} = a_{22} = 50a_0$ and $\Omega = 0.75$.

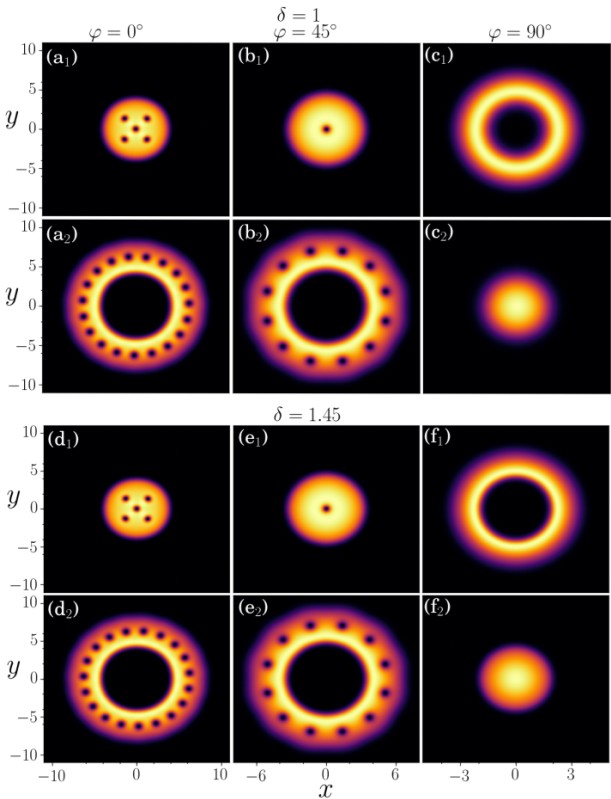

Figure 3: 2D Dipolar density patterns, $|\psi_j|^2$, where $j = 1$ is for $^{168}$Er and $j = 2$ for $^{164}$Dy, are shown for $\delta = 1$ [$(a_j)$ to $(c_j)$] and $\delta = 1.45$ [$(d_j)$ to $(f_j)$]. The dipole polarization angles ($\varphi = 0°$, $45°$, $90°$) are indicated at the top of each column, with the $^{168}$Er component having the addition of a quartic trap with $\kappa = 0.05$. The other parameters are: $N_{j=1,2} = 5000$, $\lambda = 20$, $a_{11} = a_{22} = 50a_0$ and $\Omega = 0.75$.

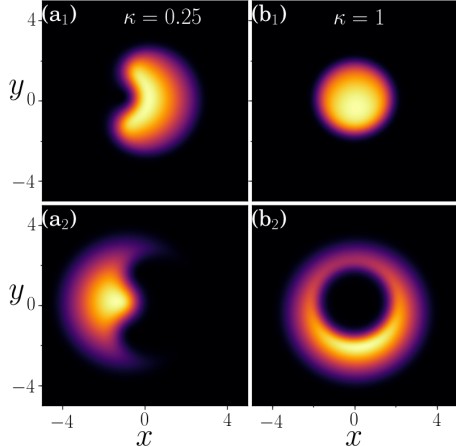

Figure 4: 2D Dipolar density patterns, $|\psi_j|^2$, where $j = 1$ is for $^{168}$Er and $j = 2$ for $^{164}$Dy, are shown for $\varphi = 90°$ and $\delta = 1.45$. The quartic trap added to component 1 is such that $\kappa = 0.25$ in the left panel and 1 in the right panel. As in Fig. 3, the other parameters are: $N_{j=1,2} = 5000$, $\lambda = 20$, $a_{11} = a_{22} = 50a_0$ and $\Omega = 0.75$.

erties, miscibility aspects, and spatial separation of these two coupled binary BEC systems, by analyzing an investigation previously reported in Ref. [12]. In addition, new results are pre-

sented by considering one of the elements of the coupled mixture being confined by a quartic interaction, which is added to the previous harmonic trap potential. The relevance of this study relies on current experimental possibilities in cold-atom laboratories to investigate such dipolar binary systems. The stability regime and miscibility properties due to the DDI of the coupled system are obtained numerically, by solving the corresponding GP equation within a model where the mixture is first confined by strong pancake-like harmonic-trap potential with aspect-ratio $\lambda = 20$ and considering repulsive two-body interactions. The DDI are tuned from repulsive to attractive by varying the dipole polarization angle, with a clear spatial separation verified in the densities for attractive DDI, being angular for symmetric mixtures and radial for asymmetric ones in the case that no quartic term is present. In an extension of our previous reported work, by adding the quartic term to the trap interaction, here we show how the density distribution of both binary system, symmetric and asymmetric ones, are affected. As shown, the quartic trap supports radial phase separations with ring lattice for both $^{164}$Dy-$^{162}$Dy and $^{168}$Er-$^{164}$Dy BEC mixtures, modifying the previous vortex-pattern structures and spatial separations obtained without the quartic term interaction. Even a weak quartic trap is enough to modify the angular spatial separation to radial ones in the dipolar $^{164}$Dy-$^{162}$Dy mixture, for attractive dipolar interactions. In the asymmetric $^{168}$Er-$^{164}$Dy dipolar BEC mixture, where we have already radial spatial separation for attractive dipolar interactions even without the quartic term, with the $^{168}$Er element surrounding the other element, we have observed that, for the addition of enough large quartic term to the $^{168}$Er element, there is an exchange of the two coupled densities, with this element moving to the center. So, for asymmetric mixture with repulsive inter-species interaction and attractive DDI, strong quartic trap ($\kappa \geq 1$) will prevent exchanges between both densities, which will remain completely radial-separated spatially.

## Acknowledgements

The authors acknowledge partial support received from Conselho Nacional de Desenvolvimento Científico e Tecnológico (CNPq) [LT, RKK and AG], Fundação de Amparo à Pesquisa do Estado de São Paulo (FAPESP) [Contracts 2016/14120-6 (LT), 2016/17612-7 (AG)]. RKK also acknowledge support from Marsden Fund (Contract UOO1726).

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
