# Peer review of "Dipolar condensed atomic mixtures and miscibility under rotation"

_SciPost Physics Proceedings, doi:SciPost Phys. Proc. 3, 023 (2020)_

## Round 1 · Referee Report · Anonymous (Referee 1) · 2019-11-22

Strengths

  1. The study of ultracold dipolar mixtures, in particular dipolar condensates, is a timely problem, and this contribution points to interesting options to control the miscibility/overlap of such two-component systems.
  2. The variety of parameters explored provide a broad overview of the options offered by theses systems, with realistic parameters, and a relevant choice of atomic species.
  3. The problem, methods and results are presented in a succinct and clear manner.

Weaknesses

  1. There is not much discussion about how the quartic potential could be created in practice, and why it would affect only (or mostly) the heavier species. While for the Er-Dy mixture it is clear that this may help invert the inside-outside densities, for the Dy-Dy mixture the situation is less clear. If this additional trapping energy is considered to be due to the same trapping laser creating the harmonic confinement, it would appear that it should affect the lighter species more, as this is the one that will generally explore a larger spatial region.

Report

This manuscript presents a detailed numerical study of the miscibility properties of two-component dipolar condensates in both harmonic and harmonic + quartic potentials, under rotation.

Specifically, the authors have considered mixtures of two dysprosium isotopes, or a dysprosium isotope with an erbium isotope. The same authors have recently reported on the distinct miscibility properties of these two mixtures due to the differing role of contact interactions, which lead to a broad variety of density profiles for the two systems as a function of rotation velocity and of the strength of the dipole-dipole interaction. Here, they show how these density profiles can be modified by adding a quartic potential that affects one of the two species, namely the heavier one in each case.

The main finding in the present manuscript is the possibility to affect the pattern of density distribution in both 'symmetric' and 'asymmetric' mixtures by controlling the additional quartic potential. In particular, the authors show it is possible to (i) induce the lead to radial phase separation [in particular, the appearance of the 'ring lattice'] in the Dy-Dy mixture [Fig. 1], and (ii) reverse the inner-outer density imbalance in the Er-Dy mixtures [Figs. 3 and 4], in contrast to the observations in their earlier work without quartic potential. These suggestions may be of interest to ongoing experimental efforts.

Because of the timeliness of this work, together with the broad range of parameters studied with well-established numerical techniques, I recommend publication of this manuscript in SciPost Physics Proceedings.

Requested changes

  1. The authors should introduce earlier on what they mean by "symmetric" and "asymmetric" mixtures. This is done only on page 5 (lines 166-167), when the concepts have been used much earlier in the text.
  2. The authors should provide some statement on the potential way to generate a controllable quartic potential in practice (e.g., an additional laser detuned from a particular transition of one species, so as to affect only that one?).
  3. The authors could indicate the advantages, if any, of calculating the dipole-interaction term by moving to momentum (Fourier) space. Does this provide any numerical advantage?

---

## Round 3 · Referee Report · Anonymous · 2019-12-5

Report

The authors have addressed the main requests raised in the earlier referee report, with the only exception of a motivation for moving to Fourier space for their numerics. I do not find this a critical point questioning the validity of the paper, and thus I see it fit for publication in SciPost Physics Proceedings.

---

## Round 3 · Author Response

Response to the Referee's report

We thank the Referee for the positive evaluation of our study, as well as for recommendations which helped us to improve the manuscript. Below, we summarize the changes made in our revision, following the Referee's comments.

Referee:

Weaknesses: There is not much discussion about how the quartic potential could be created in practice, and why it would affect only (or mostly) the heavier species. While for the Er-Dy mixture it is clear that this may help invert the inside-outside densities, for the Dy-Dy mixture the situation is less clear. If this additional trapping energy is considered to be due to the same trapping laser creating the harmonic confinement, it would appear that it should affect the lighter species more, as this is the one that will generally explore a larger spatial region.

Authors: To address this suggestion, we have included [just after the Eq.(3)] a discussion about the experimental implementation of quartic potential, in which we cite a new reference (see [13]). Details are given in section 4 of this reference, as well as in other within cited references. We should further clarify here that both elements of the mixture can be trapped independently, with the quartic potential affecting only one of the components, chosen to be the heavier mass one. For this choice, by following details given in our Ref.[12], we need to remind that the mass ratio between the components affects the miscibility properties; and that the inside-outside positions of the densities (when going from zero to 90 degrees) in case of Er-Dy mixture is due to anisotropic effects of the dipolar interaction (even before adding the quartic potential).
Close to 90 degrees, Dy is providing more attractive interaction than Er.
Therefore, the radius of the Dy density shrinks, with the Er being pulled out at large $\varphi$. The quartic interaction with strength $\kappa$=0.05 is not enough to invert their radial-spacial positions. For that, we need to increase the strength $\kappa$ of the quartic interactions, as verify by the Fig.4.

Referee: 1. The authors should introduce earlier on what they mean by symmetric" andasymmetric" mixtures. This is done only on page 5 (lines 166-167), when the concepts have been used much earlier in the text.

Authors: To address this comment, in our revision of the subsection ``Parametrization and numerical approach", we use the first sentences to clarify these terms, as being related to the dipolar symmetry of the mixtures. In the abstract and also along the text we try to emphasize that, by using "symmetric" and "asymmetric" as adjective for dipolar (as one could also think about the mass symmetry).

Referee: 2. The authors should provide some statement on the potential way to generate a controllable quartic potential in practice (e.g., an additional laser detuned from a particular transition of one species, so as to affect only that one?).

Authors: In the revised version, we have included a reference [13] related to the experimental possibilities for using quartic potentials in BECs. Within a mixture of two species, one should select one of them to apply the laser detuned for quadratic-quartic interaction.

Referee: The authors could indicate the advantages, if any, of calculating the dipole-interaction term by moving to momentum (Fourier) space. Does this provide any numerical advantage?

Authors: The non-local term with the magnetic-dipolar interactions in the integrand of Eq.(2) is quite singular at the origin, having non-trivial numerical solution in coordinate space. In view of that, one can circumvent this issue, by using Fourier convolution, which is of easy numerical implementation, as shown with more details in Ref.[10].

---

## Round 3 · List of Changes

[1-] In the subsection 2.1, ``Model formalism", a discussion about the quartic trap and possible experimental realization is included within the seven lines after Eq.(3). There, we have also pointed out a reference [13] where more details can be found.

[2-] In the subsection 2.2, ``Parametrization and numerical approach", the first four lines are new, by clarifying our ``symmetric" and ``asymmetric" definitions for the particular mixtures we are considering. Also, along the text we did some small modifications, accordingly (emphasizing the symmetry properties as adjective to dipolar).

[3-] We remove the previous Ref.[13], because we notice that it was not being cited in the manuscript. In practice it was replaced by the new reference [13], which is being used to indicate possible experimental realizations.

---

## Editorial Decision

published